# Clinical improvement in canine pulmonary hypertension with *Perna canaliculus* oil (PCSO-524) add-on therapy: Effects on exercise tolerance and cough

Nattapon Riengvirodkij[1], Mookmanee Taechikantaphat[2], Pichayut Ampapol[3], Theethach Kovorakul[3], Sapon Intaranat[3], Nick Costa[4], Walasinee Sakcamduang[2]*

1 Prasu Arthorn Veterinary Teaching Hospital, Faculty of Veterinary Science, Mahidol University, Nakhon Pathom, Thailand, 2 Department of Clinical Sciences and Public Health, Faculty of Veterinary Science, Mahidol University, Nakhon Pathom, Thailand, 3 Faculty of Veterinary Science, Mahidol University, Nakhon Pathom, Thailand, 4 College of Science, Health, Engineering and Education, Murdoch University, Murdoch, Australia

* walasinee.sak@mahidol.ac.th

## Abstract

Pulmonary hypertension (PH) in dogs, characterized by abnormally high blood pressure in the pulmonary arteries, presents a management challenge, and adjunctive therapies for the standard treatment of sildenafil are needed. This study aimed to determine whether the nutraceutical *Perna canaliculus* oil (PCSO-524), a marine lipid extract with anti-inflammatory properties, provides additional clinical benefits to dogs with PH. In a prospective, randomized, placebo-controlled trial, seventeen client-owned dogs diagnosed with PH were assigned to receive either PCSO-524 or a placebo as an add-on to their sildenafil-based therapy for 70 days. Key outcomes were evaluated using owner-assessed clinical scores for exercise tolerance and coughing, alongside echocardiographic measurements. The PCSO-524 group demonstrated a statistically significant improvement in exercise tolerance scores by day 70 (P = 0.009). This group also showed a greater reduction in coughing scores compared to the placebo group between day 35 and day 70 (P = 0.03). While the tricuspid regurgitation pressure gradient (TRPG), an estimate of pulmonary pressure, decreased significantly over time in all dogs (P = 0.001), no significant difference was found between the groups for this specific measure. These results indicate that PCSO-524 can serve as a beneficial adjunctive therapy for improving important clinical signs, such as exercise tolerance and coughing, in dogs with PH. This suggests it may be a valuable addition to standard management for enhancing quality of life.

**Data availability statement:** All relevant data are within the manuscript and its Supporting information files.

**Funding:** This study was funded by the Pharmalink International Ltd. Central, Hong Kong. The Funder had no role in study design, data collection and analysis, decision to publish, or preparation of the manuscript.

**Competing interests:** The authors have declared that no competing interests exist.

## Introduction

Pulmonary hypertension (PH) is a pathological condition defined by elevated pressure in the pulmonary arteries, most commonly occurring secondary to advanced myxomatous mitral valve disease (MMVD) in dogs [1–3]. The management of affected dogs is complex, as they often suffer from debilitating clinical signs such as coughing, exercise intolerance, dyspnea, and syncope. Assessing the disease and response to treatment is therefore multifaceted, relying on tracking these clinical outcomes, owner-assessed quality of life, and circulating biomarkers like amino-terminal pro-brain natriuretic peptide (NT-proBNP) [4,5]. While the current standard of care includes sildenafil to reduce pulmonary arterial pressure [6]. However, while sildenafil targets the hemodynamic component of the disease, many dogs continue to experience persistent clinical signs. This leaves a clear clinical need for effective adjunctive therapies aimed at further improving patient well-being and these key outcomes.

Growing evidence suggests that inflammation plays a key role in the pathophysiology and progression of both underlying cardiac disease and PH itself [6–8]. Therefore, targeting inflammation represents a promising therapeutic strategy.

*Perna canaliculus* oil (PCSO-524), a stabilized lipid extract from the New Zealand green-lipped mussel, is a nutritional supplement recognized for its potent anti-inflammatory properties [9,10]. Its effects are primarily attributed to a unique profile of omega-3 fatty acids that modulate key inflammatory pathways, such as those involving cyclooxygenase and lipoxygenase [11,12]. PCSO-524 has been shown to be safe and effective for managing chronic inflammatory conditions like osteoarthritis in both dogs and humans [13–16], but its potential benefits for cardiopulmonary diseases remain unexplored.

Given the inflammatory component of PH and the established anti-inflammatory action of PCSO-**524**, this study was designed to investigate its utility as an add-on therapy. The primary objective was to conduct a prospective, randomized, placebo-controlled trial to evaluate whether PCSO-**524** supplementation improves clinical outcomes, echocardiographic parameters, and quality of life in dogs with PH receiving standard sildenafil treatment. We hypothesized that dogs receiving PCSO-524 would demonstrate greater improvement in clinical signs compared to those receiving a placebo.

## Materials and methods

The study protocol was reviewed and approved by the Faculty of Veterinary Science – Animal Care and Use Committee (Approval No. MUVS-2018-06-32). All methods were performed in accordance with relevant guidelines and regulations. Prior to enrollment at Prasu Arthorn Veterinary Teaching Hospital, Faculty of Veterinary Science, Mahidol University, written informed consent was obtained from the owners of all participating client-owned dogs. The study was prospective conducted from October 2018 to April 2020. Dogs were required to be at least 5 years of age with no limit of sex, breed, or body weight for inclusion into the study, and informed consent were obtained from the owners. History taking, physical examination, hematological

test and serum chemistry evaluation were carried out to exclude other systemic disorders, such as hepatic disease, renal disease and infectious disease, because of metabolic interference of drug treatments. All dogs were tested for heartworm disease and excluded if positive. Furthermore, to avoid confounding analgesic effects on owner-assessed clinical scores, dogs receiving specific ongoing medication for orthopedic disease, such as non-steroidal anti-inflammatory drugs (NSAIDs), were also excluded from enrollment. In addition, tricuspid regurgitation (TR) flow velocity was measured via echocardiography. Dogs with TR flow velocity > 3.0 m/s with or without the anatomic site of echocardiographic signs of PH were classified as intermediate or high probability of PH [6] and were enrolled into the study. Seven ml of blood was collected from each dog via the lateral saphenous vein or cephalic vein using 22G needle at day 0 (visit 1) before treatment, and day 28 (visit 2), and day 56 (visit 3) post enrolment and treatment. The blood that was collected was separated into two parts; 0.5 ml into a EDTA tube for hematology by Animal Blood Counter ABCTM (Horiba ABX Diagnostic Ltd., Bangkok, Thailand), the remaining fraction into a plain tube then centrifuged to separate the serum for serum biochemistry by a Sapphire 400 Auto-Chemistry Analyzer (D.A.P. Siam Group Ltd, Bangkok, Thailand) and canine NT-proBNP by a Vcheck analyzer (Bionote Inc., Korea) [17–19].

One well-trained investigator (NR) performed the echocardiographic examination using a GE Vivid E9 with a multi-frequency sector transducer (4.5–12 MHz) with continuous electrocardiography recording. The enrolled dogs underwent the procedure without sedation, and all measurements were repeated for at least three consecutive cardiac cycles. The right parasternal long-axis view on 2-dimensional echocardiography was operated to assess the right ventricle and atrial size, the valve structure and function, including valve degeneration, valvular prolapse, and chordae tendineae. Color flow Doppler was used to identify the presence of valvular regurgitation. The left atrial to aorta ratio was measured during diastole in the right parasternal short-axis 2-dimensional view to identify left atrial dilatation [20]. The main pulmonary artery was also assessed in the right parasternal short-axis 2-dimensional view. Short-axis M-mode echocardiographic examination of the left ventricle was used to display the percentage of fractional shortening, the left ventricular internal diameter during diastole, and the left ventricular internal diameter during systole. To normalize these values based on body weight, the left ventricular internal diameter during diastole (cm) was divided by (body weight (kg))$^{0.294}$ resulting in the normalized left ventricular internal diameter in diastole, and the left ventricular internal diameter during systole (cm) was divided by (body weight (kg))$^{0.315}$ to obtain the normalized left ventricular internal diameter in systole, respectively [21]. The left parasternal views of the apical 4-chamber, the long-axis view of theFF right auricle, and the cranial transverse view of the tricuspid valve that yielded the optimal alignment of the TR flow were used to measure the maximal flow velocity of the TR. Peak TR was calculated the systolic PG across the tricuspid valve, representing the systolic pulmonary arterial pressure using the modified Bernoulli equation (pressure gradient = 4 x velocity2. Pulmonary stenosis was excluded before the diagnosis of PH [22]. PH was diagnosed based on Doppler echocardiographic finding of TR flow velocity greater than 3.0 m/s as mentioned above. Echocardiographic signs of thickened or prolapsed mitral leaflets with indications of colour-flow mitral regurgitation were used to make the diagnosis of MMVD. Dogs with MMVD were staged according to the American College of Veterinary Internal Medicine standard to determine their severity. Stage B2 was identified as having cardiomegaly on echocardiography, including a left atrial to aorta ratio greater than 1.6 and a normalized left ventricular internal diameter in diastole greater than 1.7, but not having any clinical indications of congestive heart failure (CHF). Stage C was identified by the existence of CHF symptoms and the above-mentioned stage B2 cardiac enlargement symptoms [7].

Electrocardiography was recorded using a standard six-lead recording system in right lateral recumbency. Six-lead electrocardiographic data were recorded for approximately 10 seconds, then a further one-minute recording of lead II with the paper speed set at 25 mm/s was performed for subsequent calculating heart rate and vasovagal tonus index [23,24].

Dogs enrolled in the study were randomly assigned to one of two treatment groups: the PCSO-524 group or the placebo group. All dogs received their assigned supplement (either PCSO-524 or an identical placebo) in addition to their conventional treatments, including sildenafil at a dose of 1 mg/kg orally every 12 hours. The severity of PH was used as a blocking factor in the randomization process to ensure an equal distribution of severe cases between the groups. The

randomization process was stratified using PH severity as a blocking factor to ensure a balanced distribution of severe cases. For each severity block, a unique randomization sequence was generated using a computer-based random number generator. Allocation was concealed using sequentially numbered, opaque, sealed envelopes prepared according to this sequence. To ensure blinding, the envelopes were labeled neutrally as containing either 'Medication A' or 'Medication B', which corresponded to the PCSO-524 and placebo capsules. Placebos were manufactured to be identical in appearance to PCSO-524. Sildenafil was administered by the time of initiation PCSO-524 or placebo. PCSO-524 or placebo was added according to the recommendation, loading dose at 1 soft gel capsule per dog twice a day for 2 weeks, then 1 soft gel capsule once a day for the rest of the study. Investigators, owners, and sponsors were blinded to treatment allocation. Unblinding was performed following analysis.

Clinical evaluation was assessed at entry with a questionnaire by the dogs' owners. Most of them filled out the questionnaires on their own without help from the investigators. Except for some owners who have some problems in reading (e.g., elderly owners, visually impaired owners), the investigators will help read the questions and ask for a reply. Clinical signs affecting the quality of life were scored according to the modified Häggström's et al (2008) [25], if the dog showed very good signs, the dog was assigned to score 1, but dogs with poor signs were assigned to score 3 or 4, as shown in S1 Table. All dogs enrolled in the study were scheduled for re-examination for visit 2 and visit 3 on day 28 and day 56, respectively, and all clinical procedures were performed. The follow-up intervals of day 28 and day 56 were selected based on a combination of clinical and practical considerations. A one-month follow-up (day 28) represents a standard re-examination point for chronic conditions and was considered a sufficient period to observe potential initial effects from the nutraceutical supplement. The second follow-up at day 56 was planned to assess the sustained effects of the intervention over a more prolonged period.

Computerized statistical software (IBM SPSS version 29 for Windows, Chicago, IL, USA) was used for analyses. A p-value (P) < 0.05 was considered statistically significant. The numerical parameters obtained from each group of each day of examination were tested for normality by the Shapilo- Wilk test. Comparisons of data sets shown to have a normal distributed were undertaken by independent T-test or repeated measure general linear model as appropriate. Comparisons of data sets that were not normally distributed or ordinal variables were undertaken by Mann- Whitney U test or Friedman test as appropriate. For comparison of clinical variables scores between groups at specific visits, dogs in each group were categorized as improved, unchanged, or deteriorated and tested by the Chi-square test. Results are reported as mean ± standard deviation for normally distributed variables and median with interquartile ranges for non-normally distributed variables.

All protocols in this study were approved by the Faculty of Veterinary Science, Mahidol University-Institute Animal Care and Use Committee, Thailand (COA No. MUVS-2018-06-32), in compliance with the Animals for Scientific Purposes Act, B.E. 2558 (A.D. 2015), Kingdom of Thailand. Written informed consent was obtained from the owners for the participation of their animals in this study.

## Results

### Study population and baseline characteristics

Twenty-two dogs were initially enrolled, and seventeen dogs completed the 56-day study: eight in the PCSO-524 group and nine in the placebo group. The PCSO-524 group consisted of 4 Miniature Poodles, 2 Shih Tzus, and one each of Yorkshire terrier, and Pomeranian. The placebo group included 4 Miniature Poodles, 3 Chihuahuas, and one each of Shih Tzu, and mixed. Five dogs were excluded during the study: 4 from the PCSO-524 group (3 lost to follow-up, and 1 non-compliance), and 1 from the placebo group (lost to follow up). A flow diagram outlining the number of dogs in each group during the study is depicted (Fig 1).

The baseline characteristics (day 0) of the two groups are compared in Tables 1–3, and S3 Table. There were no significant differences between the groups for all baseline variables, with two exceptions: plasma protein was significantly

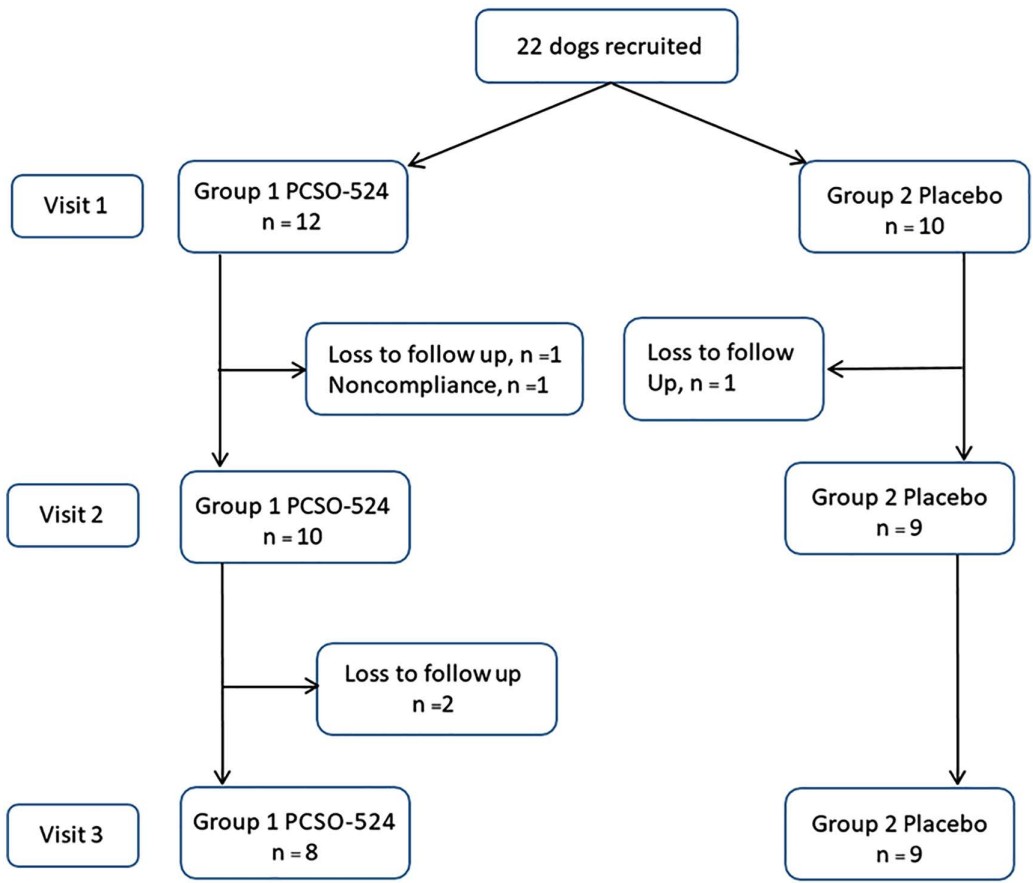

**Fig 1. A flow diagram outlining the number of dogs in each group during the study (visit 1 = day 0, visit 2 = day 35, and visit 3 = day 70).**

**Table 1. The baseline characteristics of classifications and body condition score of the 2 groups (frequencies) at entry (visit 1, day 0).**

| Variables | | PCSO-524 (n = 8) | Placebo (n = 9) | P value |
|---|---|---|---|---|
| Sex | Male | 6 | 4 | 0.20 |
| | Female | 2 | 5 | |
| ACVIM classification | B2 | 2 | 1 | 0.32 |
| | C | 6 | 6 | |
| | D | 0 | 2 | |
| Body condition score | 4/9 | 4 | 1 | 0.15 |
| | 5/9 | 3 | 7 | |
| | 8/9 | 0 | 1 | |
| | 9/9 | 1 | 0 | |

ACVIM, The American College of Veterinary Internal Medicine.

**Table 2. Distribution of continuous variables of all 3 visits (visit 1 = day 0, visit 2 = day 35, and visit 3 = day 70) of the PCSO-524 group and the placebo group. Results are shown as mean ± standard deviation (SD) for normally distributed variables and median with interquartile ranges for non-normally distributed.**

| Variables/visits | PCSO-524 (n = 8) | Placebo (n = 9) | P (between PCSO-524 and placebo) | P (repeated measured PCSO-524) | P (repeated measured placebo) |
|---|---|---|---|---|---|
| Age (years) | 12.1 ± 2.1 | 12.1 ± 3.9 | 0.97 | | |
| Body weight (kg) | | | | 0.34 | 0.99 |
| Day 0 | 5.0 (3.7, 6.3) | 5.0 (3.9, 6.8) | 0.76 | | |
| Day 35 | 4.4 (3.9, 6.2) | 5.3 (3.7, 6.8) | 0.70 | | |
| Day 70 | 4.5 (3.6, 5.9) | 5.4 (3.5, 7.0) | 0.53 | | |
| Body temperature (F) | | | | 0.30 | 0.59 |
| Visit 1 | 100.7 ± 0.9 | 101.1 ± 0.6 | 0.32 | | |
| Day 35 | 101.2 ± 0.7 | 100.9 ± 0.8 | 0.55 | | |
| Day 70 | 100.8 ± 0.7 | 101.3 ± 0.9 | 0.18 | | |
| HR-PE (bpm) | | | | 0.48 | 0.71 |
| Day 0 | 136 (123, 143) | 140 (134, 140) | 0.61 | | |
| Day 35 | 124 (120, 140) | 140 (124, 150) | 0.18 | | |
| Day 70 | 120 (111, 135) | 140 (120, 175) | 0.05 | | |
| Respiratory rate (BrPM) | | | | 0.53 | 0.40 |
| Day 0 | 29 (24, 34) | 25 (24, 47) | 0.77 | | |
| Day 35 | 28 (25, 31) | 25 (23, 33) | 0.70 | | |
| Day 70 | 26 (24, 31) | 28 (23, 31) | 0.84 | | |
| SBP (mmHg) | | | | 0.53 | 0.31 |
| Day 0 | 154 (142, 175) | 147 (138, 194) | 0.92 | | |
| Day 35 | 162 (130, 187) | 157 (144, 216) | 0.77 | | |
| Day 70 | 155 (143, 186) | 150 (146, 163) | 1.00 | | |
| VVTI | | | | 0.22 | 0.06 |
| Day 0 | 8.37 ± 2.64 | 7.40 ± 1.25 | 0.34 | | |
| Day 35 | 8.11 ± 2.06 | 8.41 ± 1.67 | 0.74 | | |
| Day 70 | 7.68 ± 3.14 | 8.74 ± 1.28 | 0.36 | | |
| HR-ECG (bpm) | | | | 0.20 | 0.25 |
| Day 0 | 140 ± 26 | 139 ± 29 | 0.93 | | |
| Day 35 | 129 ± 17 | 125 ± 33 | 0.75 | | |
| Day 70 | 135 ± 18 | 140 ± 23 | 0.59 | | |
| LA/Ao | | | | 0.65 | 0.94 |
| Day 0 | 1.90 (1.55, 2.00) | 2.01 (1.86, 2.98) | 0.15 | | |
| Day 35 | 1.79 (1.47, 1.93) | 2.31 (1.74, 2.94) | 0.07 | | |
| Day 70 | 1.91 (1.37, 2.18) | 2.07 (1.83, 2.96) | 0.25 | | |
| LVEDDN | | | | 0.22 | 0.43 |
| Day 0 | 1.66 ± 0.44 | 1.79 ± 0.35 | 0.49 | | |
| Day 35 | 1.76 ± 0.38 | 1.76 ± 0.32 | 0.99 | | |
| Day 70 | 1.78 ± 0.44 | 1.84 ± 0.34 | 0.77 | | |
| LVESDN | | | | 0.79 | 0.80 |
| Day 0 | 0.83 ± 0.32 | 0.82 ± 0.23 | 0.93 | | |
| Day 35 | 0.85 ± 0.23 | 0.79 ± 0.24 | 0.62 | | |
| Day 70 | 0.86 ± 0.30 | 0.83 ± 0.18 | 0.84 | | |

*(Continued)*

**Table 2.** (Continued)

| Variables/visits | PCSO-524 (n = 8) | Placebo (n = 9) | P (between PCSO-524 and placebo) | P (repeated measured PCSO-524) | P (repeated measured placebo) |
|---|---|---|---|---|---|
| %FS | | | | 0.32 | 0.95 |
| Day 0 | 49.8±10.6 | 53.1±7.6 | 0.46 | | |
| Day 35 | 50.4±7.4 | 53.6±8.7 | 0.43 | | |
| Day 70 | 51.6±11.7 | 52.7±6.6 | 0.82 | | |
| EPSS (mm) | | | | 0.53 | 0.95 |
| Day 0 | 1.90 (0.22, 2.98) | 1.30 (0.35, 1.49) | 0.18 | | |
| Day 35 | 1.00 (0.20, 3.48) | 1.21 (0.50, 1.91) | 1.00 | | |
| Day 70 | 1.64 (1.32, 2.41) | 1.06 (0.80, 1.46) | 0.05 | | |
| E wave (m/s) | | | | 0.36 | 0.82 |
| Day 0 | 1.4±0.4 | 1.5±0.6 | 0.75 | | |
| Day 35 | 1.2±0.2 | 1.4±0.6 | 0.47 | | |
| Day 70 | 1.4±0.5 | 1.4±0.6 | 0.96 | | |
| A wave (m/s) | | | | 0.28 | 0.06 |
| Day 0 | 0.8±0.3 | 0.9±0.2 | 0.72 | | |
| Day 35 | 1.0±0.4 | 1.0±0.3 | 0.76 | | |
| Day 70 | 0.9±0.3 | 1.1±0.3 | 0.16 | | |
| E/A ratio | | | | 0.28 | 0.04 |
| Day 0 | 1.65 (1.06, 2.08) | 1.45 (1.21, 2.14) | 0.85 | | |
| Day 35 | 1.46 (0.76, 1.65) | 1.45 (1.17, 1.74) | 0.85 | | |
| Day 70 | 1.58 (1.05, 2.14) | 1.23 (0.98, 1.53) | 0.25 | | |
| MR (mmHg) | | | | 0.11 | 0.19 |
| Day 0 | 115.1±23.4 | 119.4±41.9 | 0.80 | | |
| Day 35 | 114.4±25.1 | 137.0±34.1 | 0.14 | | |
| Day 70 | 116.2±23.1 | 129.8±28.4 | 0.30 | | |
| TR (mmHg) | | | | 0.02 | 0.01 |
| Day 0 | 78.1±31.7[a] | 69.2±24.9[c,d] | 0.52 | | |
| Day 35 | 77.7±32.4[b] | 55.2±20.0[c] | 0.10 | | |
| Day 70 | 68.8±29.2[a,b] | 53.6±16.2[d] | 0.20 | | |
| Total protein (g/dL) | | | | 0.61 | 0.45 |
| Day 0 | 6.8±0.7 | 7.3±0.8 | 0.23 | | |
| Day 35 | 6.9±1.0 | 6.7±2.6 | 0.83 | | |
| Day 70 | 7.1±0.6 | 7.5±0.4 | 0.08 | | |
| Albumin (g/dL) | | | | 0.46 | 0.97 |
| Day 0 | 3.0 (2.5, 3.4) | 3.1 (2.8, 3.4) | 0.77 | | |
| Day 35 | 3.2 (2.9, 3.3) | 3.1 (2.8, 3.6) | 0.70 | | |
| Day 70 | 3.2 (3.1, 3.4) | 3.1 (2.9, 3.3) | 0.36 | | |
| ALP (iu/L) | | | | 0.87 | 0.57 |
| Day 0 | 118 (46, 242) | 120 (62, 242) | 0.56 | | |
| Day 35 | 106 (61, 206) | 106 (61, 206) | 0.35 | | |
| Day 70 | 123 (54, 210) | 123 (54, 210) | 0.29 | | |
| ALT (iu/L) | | | | 0.53 | 0.74 |
| Day 0 | 78 (47, 116) | 82 (42, 110) | 0.85 | | |
| Day 35 | 64 (37, 90) | 83 (54, 97) | 0.32 | | |
| Day 70 | 52 (44, 138) | 87 (32, 98) | 0.63 | | |

*(Continued)*

**Table 2.** (Continued)

| Variables/visits | PCSO-524 (n=8) | Placebo (n=9) | P (between PCSO-524 and placebo) | P (repeated measured PCSO-524) | P (repeated measured placebo) |
|---|---|---|---|---|---|
| BUN (mg/dL) | | | | 0.87 | 0.35 |
| Day 0 | 31 (18, 42) | 27 (22, 46) | 0.85 | | |
| Day 35 | 27 (18, 40) | 28 (26, 34) | 0.67 | | |
| Day 70 | 27 (21, 45) | 31 (20, 46) | 0.81 | | |
| Creatinine (mg/dL) | | | | 0.12 | 0.68 |
| Day 0 | 1.0 (0.9, 1.6) | 1.2 (0.9, 1.6) | 0.77 | | |
| Day 35 | 0.9 (0.8, 1.5) | 1.4 (1.1, 1.9) | 0.18 | | |
| Day 70 | 1.0 (0.8, 1.2) | 1.6 (1.0, 1.8) | 0.16 | | |
| RBC ($10^{12}$/L) | | | | 0.25 | 0.70 |
| Day 0 | 6.4±1.2 | 6.9±1.1 | 0.42 | | |
| Day 35 | 6.3±1.3 | 6.9±1.2 | 0.32 | | |
| Day 70 | 6.3±1.3 | 7.0±1.2 | 0.25 | | |
| PCV (%) | | | | 0.41 | 0.83 |
| Day 0 | 43.8±7.5 | 47.1±6.9 | 0.35 | | |
| Day 35 | 42.2±9.0 | 47.4±7.6 | 0.22 | | |
| Day 70 | 41.9±8.4 | 48.0±7.2 | 0.13 | | |
| Hemoglobin (g/dl) | | | | 0.58 | 0.88 |
| Day 0 | 14.6±2.3 | 16.1±2.3 | 0.20 | | |
| Day 35 | 14.2±2.8 | 16.0±2.5 | 0.17 | | |
| Day 70 | 14.2±2.7 | 16.2±2.6 | 0.14 | | |
| MCV (fL) | | | | 0.24 | 0.59 |
| Day 0 | 68.4±2.7 | 68.7±3.4 | 0.85 | | |
| Day 35 | 67.5±3.2 | 69.0±3.7 | 0.40 | | |
| Day 70 | 66.8±4.5 | 70.6±7.2 | 0.23 | | |
| MCH (pg) | | | | 0.84 | 0.37 |
| Day 0 | 22.9±1.1 | 23.5±1.3 | 0.32 | | |
| Day 35 | 22.8±1.2 | 23.4±1.5 | 0.38 | | |
| Day 70 | 22.7±1.6 | 23.1±1.3 | 0.55 | | |
| MCHC (g/dL) | | | | 0.06 | 0.49 |
| Day 0 | 33.5±0.8 | 34.4±2.0 | 0.25 | | |
| Day 35 | 33.7±0.9 | 33.8±0.7 | 0.79 | | |
| Day 70 | 34.0±0.7 | 33.8±1.1 | 0.66 | | |
| Platelet ($10^{9}$/L) | | | | 0.82 | 0.52 |
| Day 0 | 463.1±162.7 | 368.7±114.2 | 0.18 | | |
| Day 35 | 452.0±236.4 | 341.9±136.6 | 0.25 | | |
| Day 70 | 470.5±174.2 | 359.7±125.2 | 0.15 | | |
| RDW (%) | | | | 0.31 | 0.56 |
| Day 0 | 13.7±1.5 | 13.0±1.0 | 0.29 | | |
| Day 35 | 13.9±1.4 | 13.0±1.0 | 0.14 | | |
| Day 70 | 14.3±1.3 | 12.8±0.7 | 0.01 | | |
| Plasma protein (g/dL) | | | | 0.83 | 0.60 |
| Day 0 | 8.4±0.8 | 9.4±0.6 | 0.02 | | |
| Day 35 | 8.7±1.0 | 9.2±0.8 | 0.26 | | |
| Day 70 | 8.5±0.7 | 9.4±0.8 | 0.03 | | |

*(Continued)*

**Table 2.** (Continued)

| Variables/visits | PCSO-524 (n=8) | Placebo (n=9) | P (between PCSO-524 and placebo) | P (repeated measured PCSO-524) | P (repeated measured placebo) |
|---|---|---|---|---|---|
| WBC ($10^6$/L) | | | | 0.68 | 0.83 |
| Day 0 | 14,109±5,634 | 11,504±4,658 | 0.31 | | |
| Day 35 | 12,711±4,962 | 12,526±5,599 | 0.94 | | |
| Day 70 | 13,190±5,749 | 12,739±8,158 | 0.90 | | |
| Neutrophils ($10^6$/L) | | | | 0.62 | 0.87 |
| Day 0 | 11,278±4,608 | 8,695±3,996 | 0.23 | | |
| Day 35 | 9,943±4,818 | 9,172±4,186 | 0.73 | | |
| Day 70 | 10,737±5,211 | 9,676±6,881 | 0.73 | | |
| Lymphocytes ($10^6$/L) | | | | 0.72 | 0.09 |
| Day 0 | 1,898±928 | 1,876±526 | 0.95 | | |
| Day 35 | 1,851±804 | 2,688±1,450 | 0.17 | | |
| Day 70 | 1,618±411 | 2,285±1,058 | 0.12 | | |
| Eosinophils ($10^6$/L) | | | | 0.70 | 0.21 |
| Day 0 | 406±315 | 375±307 | 0.84 | | |
| Day 35 | 323±303 | 172±132 | 0.20 | | |
| Day 70 | 372±173 | 253±254 | 0.28 | | |
| Monocytes ($10^6$/L) | | | | 0.83 | 0.97 |
| Day 0 | 527±471 | 526±374 | 0.99 | | |
| Day 35 | 594±385 | 493±398 | 0.60 | | |
| Day 70 | 463±256 | 488±400 | 0.88 | | |
| Neutrophils/lymphocytes ratio | | | | 0.28 | 0.17 |
| Day 0 | 5.8 (4.8, 8.0) | 5.0 (3.2, 6.2) | 0.18 | | |
| Day 35 | 4.2 (3.5, 6.3) | 3.8 (2.5, 4.6) | 0.24 | | |
| Day 70 | 5.7 (5.1, 7.3) | 4.4 (2.7, 5.1) | 0.02 | | |
| NT-proBNP (pmol/L) | | | | 0.65 | 0.11 |
| Day 0 | 1,597 (619, 4,136) | 1,341 (899, 3,074) | 0.85 | | |
| Day 35 | 2,068 (1,013, 3,459) | 862 (484, 2,806) | 0.18 | | |
| Day 70 | 1,615 (750, 4,185) | 827 (589, 3,101) | 0.50 | | |

Comparisons of each visit between the PCSO-524 group and the placebo group were tested by independent T-test for normally distributed variables and Mann-Whitney U test for non-normally distributed variables. Comparisons between visits within subjects each group were tested by repeated measured general linear model with pairwise Dunn's test for normally distributed variables or Friedman with pairwise Wilcoxon signed-rank test for non-normal distributed variables.

[a] Statistically significant difference within subjects of the PCSO-524 group between day 0 and day 70 (P=0.02).

[b] Statistically significant difference within subjects of the PCSO-524 group between day 35 and day 70 (P=0.03).

[c] Statistically significant difference within subjects of the placebo group between day 0 and day 35 (P=0.03).

[d] Statistically significant difference within subjects of the placebo group between day 0 and day 70 (P=0.01).

HR-PE, heart rate measured from physical examination; SBP, systolic blood pressure; bpm, beats per minute; BPM, breaths per minute; VVTI, vasovagal tonus index; HR-ECG, heart rate measured from electrocardiography; LA/Ao, left atrial to aortic ratio; LVEDDN, left ventricular end-diastolic diameter normalized; LVESDN, left ventricular end-systolic diameter normalized; %FS, % fractional shortening; E to A ratio, peak of E wave to peak of A wave ratio; EPSS, E-point septal separation; MR, mitral regurgitation; TR, tricuspid regurgitation; ALP, alkaline phosphatase; ALT, alanine transferase; BUN, blood urea nitrogen; RBC, red blood cell count; PCV, packed cell volume; MCV, mean corpuscular volume; MCH, mean cell hemoglobin; MCHC, mean corpuscular hemoglobin concentration; RDW, red cell distribution width; WBC, white blood cell count; NT-proBNP, N-terminal probrain natriuretic peptide.

**Table 3. Distribution of quality of life and respiratory variables scores of all 3 visits (visit=day 0, visit 2=day 35, and visit 3=day 70) of the PCSO-524 group and the placebo group. Results are shown as median with interquartile ranges.**

| Variables/visits | PCSO-524 (n=8) | Placebo (n=9) | P (between PCSO-524 and placebo) | P (repeated measured PCSO-524) | P (repeated measured placebo) |
|---|---|---|---|---|---|
| Exercise tolerance | | | | 0.009 | 0.85 |
| Day 0 | 2 (1, 3)[a] | 2 (1, 2) | 0.50 | | |
| Day 35 | 2 (2, 3)[b] | 2 (1, 2) | 0.07 | | |
| Day 70 | 1 (1, 2)[a,b] | 2 (1, 3) | 0.27 | | |
| Demeanor | | | | 0.25 | 0.72 |
| Day 0 | 2 (1, 2) | 1 (1, 2) | 0.66 | | |
| Day 35 | 1 (1, 2) | 1 (1, 2) | 0.56 | | |
| Day 70 | 1 (1, 1) | 1 (1, 2) | 0.56 | | |
| Appetite | | | | 0.90 | 0.51 |
| Day 0 | 2 (2, 2) | 2 (2, 2) | 0.76 | | |
| Day 35 | 2 (2, 3) | 2 (2, 2) | 0.43 | | |
| Day 70 | 2 (2, 2) | 2 (2, 2) | 1.00 | | |
| Respiratory effort | | | | 0.49 | 0.82 |
| Day 0 | 2 (1, 3) | 2 (1, 2) | 0.79 | | |
| Day 35 | 2 (1, 3) | 2 (1, 2) | 0.44 | | |
| Day 70 | 2 (1, 2) | 2 (1, 2) | 0.74 | | |
| Coughing | | | | 0.82 | 0.37 |
| Day 0 | 2 (2, 2) | 2 (1, 2) | *0.03* | | |
| Day 35 | 2 (2, 3) | 2 (1, 2) | 0.07 | | |
| Day 70 | 2 (2, 2) | 2 (1, 2) | 0.21 | | |
| Noctonal dyspnea/cough | | | | 0.78 | 0.72 |
| Day 0 | 2 (1, 2) | 1 (1, 2) | 0.47 | | |
| Day 35 | 2 (1, 2) | 1 (1, 2) | 0.82 | | |
| Day 70 | 2 (1, 2) | 1 (1, 2) | 0.78 | | |

Comparisons of each visit between the PCSO-524 group and the placebo group were tested Mann-Whitney U test. Comparisons between visits within subjects each group were tested by Friedman with pairwise Wilcoxon signed-rank test.

[a] Statistically significant difference within subjects of the PCSO-524 group between day 0 and day 70 (P=0.04).

[b] Statistically significant difference within subjects of the PCSO-524 group between day 35 and day 70 (P=0.01).

higher in the placebo group (P=0.02, Table 2), and the owner-reported coughing score was significantly more severe in the PCSO-524 group (P=0.03, Table 3). The types and doses of concurrent medications were similar between groups (S2 Table), as was the prescribed daily dose of sildenafil (PCSO-524 group: 4.08±1.10 mg/kg/day (range 2.68–5.96 mg/kg/day); placebo group: 3.81±1.25 mg/kg/day) (range 2.80–6.66 mg/kg/day).

Re-examination timings were consistent between the two groups (P=0.70). The first follow-up (target day 28) occurred at a median of 35 days for both groups, with ranges of 28–39 days for the PCSO-524 group and 28–42 days for the placebo group. The second follow-up (target day 56) also occurred at a median of 35 days after the first visit, with ranges of 29–36 days for the PCSO-524 group and 28–41 days for the placebo group. Since the actual median re-examination times occurred at 35 and 70 days. Therefore, for clarity and consistency throughout the manuscript, all results are expressed using the day 0, day 35, and day 70 time points.

### Between-group comparisons at follow-up

Between-group comparisons of laboratory parameters revealed no significant differences at day 35. However, at day 70, the PCSO-524 group showed a significantly higher red cell distribution width (RDW) (P=0.01) and

neutrophil-to-lymphocyte ratio (P = 0.02), along with a lower plasma protein level (P = 0.03) compared to the placebo group (Table 2). All other parameters were not significantly different between the groups at either time point (S3 Table).

In addition to these measures, the change in owner-assessed clinical scores between follow-up visits was compared. The improvement in coughing score from day 35 to day 70 was significantly greater in the PCSO-524 group than in the placebo group (P = 0.03). A non-significant trend toward greater improvement in exercise tolerance was also observed in the PCSO-524 group during the same period (P = 0.06) (Table 4).

### Within-group changes over time

When all 17 dogs were analyzed together as a single cohort, there was a significant overall effect of time on the tricuspid regurgitation pressure gradient (TRPG) (P = 0.001), reflecting the efficacy of sildenafil therapy (S4 and S5 Tables). The mean TRPG decreased from 73.4 ± 27.8 mmHg at baseline (day 0) to 65.8 ± 28.2 mmHg at day 35, and further to 60.7 ± 23.8 mmHg at day 70 (Fig 2). Post-hoc comparisons revealed that this significantly decrease by day 35 (P = 0.01 compared to day 0) and was highly significant by day 70 (P = 0.004 compared to day 0).

Within-group analyses revealed significant changes in two key parameters over the 70-day study. The tricuspid regurgitation pressure gradient (TRPG) decreased significantly over time in both the PCSO-524 group (P = 0.02) and the placebo group (P = 0.01) (Table 2, Fig 3). For the PCSO-524 group, this reduction was statistically significant at day 70 when

**Table 4. Between the PCSO-524 group and the placebo group unpaired comparisons of the change between visits of quality of life and respiratory variables scores.**

| Variables/comparisons of visits | All dogs | | | PCSO-524 | | | Placebo | | | P (between PCSO-524 and placebo) |
|---|---|---|---|---|---|---|---|---|---|---|
| | Deteriorated (n) | Unchanged (n) | Improved (n) | Deteriorated (n) | Unchanged (n) | Improved (n) | Deteriorated (n) | Unchanged (n) | Improved (n) | |
| Exercise tolerance | | | | | | | | | | |
| Day 0 vs day 35 | 3 | 12 | 2 | 2 | 6 | 0 | 1 | 6 | 2 | 0.32 |
| Day 0 vs day 70 | 2 | 9 | 6 | 0 | 4 | 4 | 2 | 5 | 2 | 0.26 |
| Day 35 vs day 70 | 3 | 6 | 8 | 0 | 2 | 6 | 3 | 4 | 2 | 0.06 |
| Demeanor | | | | | | | | | | |
| Day 0 vs day 35 | 0 | 15 | 2 | 0 | 7 | 1 | 0 | 8 | 1 | NA |
| Day 0 vs day 70 | 2 | 10 | 5 | 1 | 3 | 4 | 1 | 7 | 1 | 0.19 |
| Day 35 vs day 70 | 3 | 10 | 4 | 1 | 4 | 3 | 2 | 6 | 1 | 0.43 |
| Appetite | | | | | | | | | | |
| Day 0 vs day 35 | 2 | 11 | 4 | 1 | 6 | 1 | 1 | 5 | 3 | 0.60 |
| Day 0 vs day 70 | 2 | 11 | 4 | 1 | 6 | 1 | 1 | 5 | 3 | 0.60 |
| Day 35 vs day 70 | 3 | 10 | 4 | 1 | 5 | 2 | 2 | 5 | 2 | 0.87 |
| Respiratory effort | | | | | | | | | | |
| Day 0 vs day 35 | 3 | 10 | 4 | 2 | 4 | 2 | 1 | 6 | 2 | 0.71 |
| Day 0 vs day 70 | 3 | 8 | 6 | 1 | 4 | 3 | 2 | 4 | 3 | 0.87 |
| Day 35 vs day 70 | 3 | 10 | 4 | 1 | 4 | 3 | 2 | 6 | 1 | 0.43 |
| Coughing | | | | | | | | | | |
| Day 0 vs day 35 | 2 | 14 | 1 | 2 | 5 | 1 | 0 | 9 | 0 | 0.13 |
| Day 0 vs day 70 | 3 | 12 | 2 | 1 | 5 | 2 | 2 | 7 | 0 | 0.27 |
| Day 35 vs day 70 | 4 | 9 | 4 | 2 | 2 | 4 | 2 | 7 | 0 | 0.03 |
| Nocturnal dyspnea/cough | | | | | | | | | | |
| Day 0 vs day 35 | 3 | 12 | 2 | 2 | 4 | 2 | 1 | 8 | 0 | 0.16 |
| Day 0 vs day 70 | 6 | 8 | 3 | 3 | 3 | 2 | 3 | 5 | 1 | 0.68 |
| Day 35 vs day 70 | 4 | 9 | 4 | 2 | 3 | 3 | 2 | 6 | 1 | 0.38 |

Different of 3 categories were tested by Chi-square and reported in column P (between the PCSO-524 group and the placebo group).

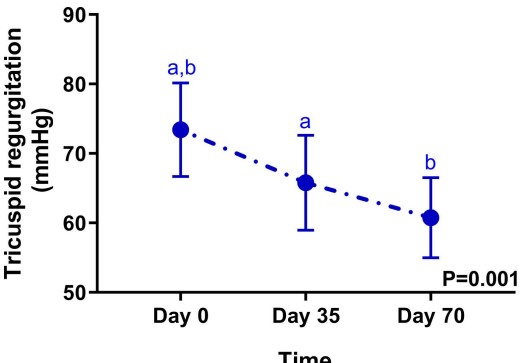

**Fig 2. Tricuspid regurgitation pressure gradient of all 17 dogs.** Repeated measured within subjects were analyzed for all 17 dogs (P=0.001). [a]Significant difference between day 0 and day 35 of all 17 dogs (P=0.01). [b]Significant difference between day 0 and day 70 of all 17 dogs (P=0.004).

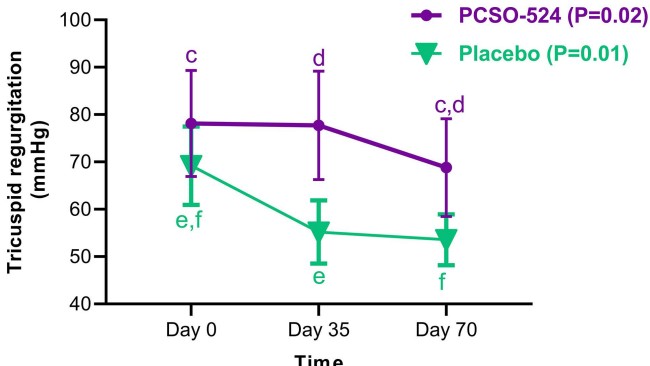

**Fig 3. Tricuspid regurgitation pressure gradient in the PCSO-524 and placebo groups.** Repeated measured within subjects were analyzed for each group of dogs (P model of the PCSO-524 group=0.02, and P model of the placebo group=0.01). [c]Significant difference between day 0 and day 70 of the PCSO-524 group (P=0.02). [d]Significant difference between day 35 and day 70 of the PCSO-524 group (P=0.03). [e]Significant difference between day 0 and day 35 of the placebo group (P=0.03). [f]Significant difference between day 0 and day 70 of the placebo group (P=0.01).

compared to both baseline (P=0.02) and day 35 (P=0.03). For the placebo group, the reduction from baseline was significant at both day 35 (P=0.03) and day 70 (P=0.01).

Notably, a significant improvement in the owner-assessed exercise tolerance score was observed exclusively in the PCSO-524 group (P=0.009). For these dogs, the score at day 70 was significantly better than at day 0 (P=0.046) and day 35 (P=0.014) (Table 3). No significant changes in exercise tolerance were observed in the placebo group.

## Discussion

This prospective, randomized, placebo-controlled study investigated the effects of PCSO-524, a marine lipid extract, as an adjunctive therapy for dogs with pulmonary hypertension (PH) receiving sildenafil. The primary finding was that dogs supplemented with PCSO-524 demonstrated statistically significant improvements in key owner-assessed clinical signs, specifically exercise tolerance and coughing, compared to dogs receiving a placebo. These results suggest that PCSO-524 can be a safe and beneficial addition to the standard management of canine PH.

The most notable clinical improvement was in exercise tolerance, which significantly improved within the PCSO-524 group over the 70-day study period. This benefit may be multifaceted. As a potent anti-inflammatory agent, PCSO-524

may have attenuated systemic or pulmonary inflammation associated with PH. However, given that many elderly, small-breed dogs have concurrent, subclinical orthopedic issues, it is also plausible that the known analgesic effects of PCSO-524 for conditions like osteoarthritis [8,12,13] improved mobility, leading to better owner-perceived exercise capacity. Further studies are required to differentiate between the cardiopulmonary and musculoskeletal benefits of the therapy.

Furthermore, the improvement in coughing score was significantly greater in the PCSO-524 group compared to the placebo group between day 35 and day 70. While the PCSO-524 group coincidentally had more severe coughing at baseline, the analysis of the change in scores supports a true treatment effect. The anti-inflammatory properties of PCSO-524 are well-documented to have beneficial effects on inflammatory airway diseases like asthma in humans [35–37], and a similar mechanism may be at play in reducing the cough associated with cardiac disease in dogs.

The clinical benefits observed in this study may also be attributable to the unique and complex composition of PCSO-524. Unlike highly purified fish oil supplements that primarily provide eicosapentaenoic acid (EPA) and docosahexaenoic acid (DHA), PCSO-524 is a stabilized whole lipid extract [26]. It contains a diverse profile of over 90 different fatty acids, including omega-3s, in combination with other lipid classes such as sterol esters and polar lipids [9]. This complex mixture may produce synergistic effects; for instance, the variety of components may inhibit multiple inflammatory pathways simultaneously, such as the cyclooxygenase pathway [12], while other lipids and natural antioxidants in the extract could enhance the stability and bioavailability of the active ingredients. Therefore, the clinical improvements seen in our study might not be due to a single component, but rather to the combined action of the entire lipid extract, distinguishing its potential effects from those of standard omega-3 supplementation.

Consistent with previous findings, sildenafil therapy was effective in reducing the tricuspid regurgitation pressure gradient (TRPG) in this study. The magnitude of this effect in our placebo group, which experienced a reduction of approximately 14 mmHg over the first follow-up period, was comparable to the 16.5 mmHg reduction reported in a prior 4-week study of dogs with PH [27]. It is notable that our results were achieved with a less frequent sildenafil dosing schedule (every 12 hours) than in the previous study. Importantly, there was no significant difference in the magnitude of TRPG reduction between the PCSO-524 and placebo groups, indicating that PCSO-524 did not have an additive pressure-lowering effect. This finding strengthens the hypothesis that the clinical benefits of PCSO-524 observed in this study are mediated through mechanisms other than vasodilation, such as its known anti-inflammatory properties.

The study population consisted primarily of elderly, small-breed dogs, with a high proportion of Miniature Poodles, reflecting a common demographic for severe myxomatous mitral valve disease (MMVD) and secondary PH in Thailand. The predominance of advanced cardiac disease (ACVIM Stage C; 82%) in our cohort is consistent with previous findings that PH risk increases with the severity of MMVD [28–30]. This is understood to be a consequence of increased left atrial pressure from severe mitral regurgitation, which raises pulmonary capillary pressure and drives the development of PH [28,31].

Regarding biomarkers, NT-proBNP concentrations did not change significantly in either group during the study, with all median values remaining consistently elevated above the 800 pmol/L reference level for cardiac disease (Table 3). We interpret this lack of change not as a treatment failure, but as a reflection of the chronic, stable nature of the underlying cardiac disease in this medically managed population. We hypothesize that the primary therapy, sildenafil, was effective in stabilizing the dogs' condition, resulting in persistently high but static NT-proBNP levels that reflect an ongoing, but no longer worsening, hemodynamic load.

To support this interpretation, the observed concentrations were notably higher than those typically reported for dogs with asymptomatic myxomatous mitral valve disease, yet lower than for those in active congestive heart failure [17–19]. This finding contrasts with reports of NT-proBNP decreasing after pimobendan treatment [4], a discrepancy we attribute to differing pharmacological actions rather than assay performance, as the assays used are comparable [32]. Therefore, our results suggest that in stable PH patients, NT-proBNP is a valuable indicator of underlying disease presence but may have limited utility for monitoring short-term therapeutic response.

Several unexpected laboratory findings at day 70 warrant a cautious interpretation. The PCSO-524 group had a significantly higher red cell distribution width (RDW) and neutrophil-to-lymphocyte (N:L) ratio, as well as a lower plasma protein level compared to the placebo group. This presents a clinical paradox regarding RDW and the N:L ratio, as these markers are typically associated with a poorer prognosis in cardiac patients [33–38], while the reason for the difference in plasma protein is unclear. However, it is crucial to note that the mean values for all these parameters remained well within their normal reference intervals, arguing against immediate clinical concern. Given the small sample size of this pilot study, these findings could be incidental. Nevertheless, the possibility that these laboratory changes could signal unknown, unintended long-term consequences cannot be dismissed. Therefore, we strongly recommend that future, larger-scale trials specifically monitor these hematological and biochemical parameters to determine their clinical significance and confirm the long-term safety profile of PCSO-524.

The present study has several limitations. The primary limitation is the small sample size (n = 17), which restricts the statistical power and the generalizability of our findings to a broader population of dogs with pulmonary hypertension. A larger study is necessary to confirm the encouraging clinical trends observed and to detect more subtle differences in other measurements. Secondly, this study relied on owner-reported scores to assess clinical efficacy. Although our scoring system was adapted from the QUEST study [39] for clinical relevance, these outcomes remain inherently subjective. Future research should therefore complement these scores with objective functional endpoints, such as the 6-minute walk test (6MWT) [40], to provide more robust evidence of therapeutic benefit. Thirdly, re-examination schedules for both groups were delayed in some cases, which could have influenced the consistency of our measurements. Finally, and perhaps most significantly, the lack of detailed dietary information for the enrolled dogs is a key limitation. The baseline intake of omega-3 fatty acids (EPA and DHA) from their regular diets was unknown and could not be accounted for. This is a potential confounding variable that may have influenced the results. Therefore, we suggest that future studies should aim to either control this variable by feeding all participants a standardized diet or, at minimum, record detailed dietary histories to allow for a more precise evaluation of the supplement's effects.

Despite these limitations, this study provides preliminary evidence that PCSO-524 is a safe add-on therapy that does not interfere with standard treatment for pulmonary hypertension. The encouraging improvements in specific clinical scores warrant further investigation in a larger, long-term trial to more definitively assess the impact of PCSO-524 on quality of life and survival in dogs with cardiovascular diseases.

## Conclusion

This study provides the first randomized, controlled evidence for the clinical utility of PCSO-524 as an adjunctive therapy for dogs with pulmonary hypertension. The addition of PCSO-524 to standard sildenafil treatment was safe, well-tolerated, and resulted in statistically significant improvements in owner-assessed exercise tolerance and coughing. While it did not offer additional reduction in echocardiographic pressure estimates, its benefits on key clinical signs suggest it is a valuable tool for improving the quality of life in canine cardiac patients. Further investigation in larger, long-term, diet-controlled trials is warranted to confirm these findings and assess the impact on survival.

## Supporting information

**S1 Table. Ordinal scoring system for quality of life and respiratory variables.**
(DOCX)

**S2 Table. Medications prescribed before administration of sildenafil, PCSO-524, and placebo.**
(DOCX)

**S3 Table. Distribution of continuous variables of all 3 visits of group 1 (placebo) and group 2 (placebo) that have no significant difference among the groups.**
(XLSX)

**S4 Table. Distribution of continuous variables of all 3 visits of all 17 dogs.**
(DOCX)

**S5 Table. Comparisons within subjects of the quality of life and respiratory variables scores of all 17 dogs (medians (interquartile)) of all visits (visit 1, 2, and 3).**
(DOCX)

**S1 Checklist. PLOSOne_Human_Subjects_Research_Checklist_250418.**
(DOCX)

## Acknowledgments

We would like to thank the owners of the dogs that willingly participated in this study; the staff at Prasu-Arthorn Veterinary Teaching Hospital, Faculty of Veterinary Science, Mahidol University, Nakhon Pathom, Thailand, for case recruitment and assisting all procedures. The work was supported by a grant from Pharmalink International Ltd. Central, Hong Kong. The NT-proBNP assays used in this study were provided free of charge by Bionote Inc., Korea, and Bestagro Co, Ltd., Thailand.

## Author contributions

**Conceptualization:** Nattapon Riengvirodkij, Mookmanee Taechikantaphat, Walasinee Sakcamduang.

**Data curation:** Nattapon Riengvirodkij, Mookmanee Taechikantaphat, Walasinee Sakcamduang.

**Formal analysis:** Nattapon Riengvirodkij, Mookmanee Taechikantaphat, Walasinee Sakcamduang.

**Funding acquisition:** Walasinee Sakcamduang.

**Investigation:** Nattapon Riengvirodkij, Mookmanee Taechikantaphat, Pichayut Ampapol, Theethach Kovorakul, Sapon Intaranat, Walasinee Sakcamduang.

**Methodology:** Nattapon Riengvirodkij, Mookmanee Taechikantaphat, Pichayut Ampapol, Theethach Kovorakul, Sapon Intaranat, Walasinee Sakcamduang.

**Project administration:** Walasinee Sakcamduang.

**Resources:** Nattapon Riengvirodkij, Mookmanee Taechikantaphat, Walasinee Sakcamduang.

**Software:** Walasinee Sakcamduang.

**Supervision:** Walasinee Sakcamduang.

**Validation:** Nattapon Riengvirodkij, Mookmanee Taechikantaphat, Walasinee Sakcamduang.

**Visualization:** Nattapon Riengvirodkij, Mookmanee Taechikantaphat, Walasinee Sakcamduang.

**Writing – original draft:** Nattapon Riengvirodkij, Mookmanee Taechikantaphat, Walasinee Sakcamduang.

**Writing – review & editing:** Nattapon Riengvirodkij, Mookmanee Taechikantaphat, Pichayut Ampapol, Theethach Kovorakul, Sapon Intaranat, Nick Costa, Walasinee Sakcamduang.

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
