## [Decision Letter · Decision Letter 0]

4 Jun 2025

Dear Dr. Sakcamduang,

Thank you for submitting your manuscript to PLOS ONE. After careful consideration, we feel that it has merit but does not fully meet PLOS ONE’s publication criteria as it currently stands. Therefore, we invite you to submit a revised version of the manuscript that addresses the points raised during the review process.

We look forward to receiving your revised manuscript.

Kind regards,

Doa'a G. F. Al-u'datt

Academic Editor

PLOS ONE

Journal Requirements:

2. Thank you for your submission to PLOS ONE. We note that your cover letter mentions an article with the title "Effect of Perna canaliculus oil in dogs with pulmonary hypertension", but the main manuscript's title is "Clinical Improvement in Canine Pulmonary Hypertension with Perna canaliculus Oil (PCSO-524) Add-on Therapy: Effects on Exercise Tolerance and Cough". Can you please clarify which is the correct title and check if you have submitted an incorrect cover letter or incorrect manuscript file? If one is incorrect please upload the correct file.

This study was funded by the Pharmalink International Ltd. Central, Hong Kong.

4. Please remove all personal information, ensure that the data shared are in accordance with participant consent, and re-upload a fully anonymized data set.

Reviewers' comments:

Reviewer's Responses to Questions

**Comments to the Author**

1. Is the manuscript technically sound, and do the data support the conclusions?

Reviewer #1: Partly

Reviewer #2: Yes

Reviewer #3: Partly

Reviewer #4: Yes

Reviewer #5: Yes

Reviewer #6: Partly

Reviewer #7: Yes

Reviewer #8: Partly

2. Has the statistical analysis been performed appropriately and rigorously?

Reviewer #1: Yes

Reviewer #2: Yes

Reviewer #3: Yes

Reviewer #4: Yes

Reviewer #5: Yes

Reviewer #6: Yes

Reviewer #7: Yes

Reviewer #8: Yes

3. Have the authors made all data underlying the findings in their manuscript fully available?

Reviewer #1: Yes

Reviewer #2: Yes

Reviewer #3: Yes

Reviewer #4: Yes

Reviewer #5: Yes

Reviewer #6: Yes

Reviewer #7: Yes

Reviewer #8: Yes

4. Is the manuscript presented in an intelligible fashion and written in standard English?

Reviewer #1: Yes

Reviewer #2: Yes

Reviewer #3: No

Reviewer #4: Yes

Reviewer #5: Yes

Reviewer #6: No

Reviewer #7: Yes

Reviewer #8: Yes

Reviewer #1: This is an interesting and well-designed study that evaluates the potential benefits of Perna canaliculus oil (PCSO-524) as an add-on therapy in dogs with pulmonary hypertension. The topic is relevant to clinical veterinary medicine, especially for practitioners managing canine cardiac patients.

Strengths of the manuscript:

The prospective, randomized, placebo-controlled trial design adds scientific strength and reliability to the findings.

The methods, particularly the use of standardized clinical scoring and echocardiographic assessment, are appropriately selected.

The writing is generally clear and well-organized.

Ethical approval and owner consent were obtained and clearly reported.

Suggestions for improvement:

1. Sample size: The study had a relatively small number of dogs (n=17), which limits the generalizability of the findings. Please acknowledge this limitation more clearly in the discussion.

2. Dietary influence: Since omega-3 intake from the dogs' diets may have affected the results, future studies should aim to control or record dietary intake.

3. Biomarker use: The lack of significant change in NT-proBNP concentrations deserves further exploration or explanation, especially considering its role in heart disease monitoring.

4. Clarity of figures and tables: Ensure that all supplemental tables and figures referenced (e.g., Supplemental Table 3, Figure 1) are available and labeled clearly for reviewers and readers.

5. Minor language editing: Some grammatical and typographical errors should be addressed to improve the overall clarity and professionalism of the manuscript.

Ethical and scientific integrity: There are no major concerns regarding ethical issues, dual publication, or research misconduct. The study appears to adhere to the ethical standards of animal research.

In conclusion, this study contributes valuable preliminary data on the potential use of PCSO-524 in dogs with PH. With minor revisions and clarifications, the manuscript could be suitable for publication.

Reviewer #2: The specific questions and recommendations are as follows:

1:

The manuscript would benefit from a structured abstract.

2:

It is recommended to crate a section for the Ethical considerations. Under Ethical considerations

section, it is required to provide more information in the standard format.

3:

I recommend creating a new section for the study limitations.

4:

It is advised that a new section explaining any acronyms used in the article be added at the end,

following the conclusion section.

Thanks to the authors for their hard work on this Study. It’s a pleasure to review such a

contribution to the field.

Reviewer #3: Abstract : "While the abstract contains the necessary scientific content, it currently lacks the clarity and brevity expected by PLOS ONE. The sentences are overly long and dense, obscuring the main findings. I recommend reducing redundancy, improving flow, and focusing on clear communication of the objective, methods, and outcomes in under 300 words." Introduction :

The Introduction provides a reasonable overview of pulmonary hypertension (PH) in dogs, including its causes, clinical manifestations, and current treatment options such as sildenafil. It also introduces PCSO-524 and its known anti-inflammatory properties. However, the section could be improved in several ways. The transitions between ideas are somewhat abrupt, and the research gap—why adjunctive therapy is needed despite standard treatment—is not clearly articulated. The mechanistic discussion of PCSO-524's effects is overly detailed and would benefit from being more concise and clinically focused. Additionally, there are several grammatical and formatting issues (e.g., “outcomesin” in line 75) that require correction. I recommend clarifying the rationale for the study and tightening the prose to improve readability and focus.

Materials and Methods

The methodology is overall well-designed and clearly reported. The study is a prospective, randomized, placebo-controlled trial with appropriate ethical approvals and informed consent. Interventions and clinical assessments, including echocardiography, electrocardiography, and biomarker analysis, are detailed and replicable. The inclusion criteria and statistical analysis are also appropriate. Blinding is reported and appears to be well-executed. However, additional clarification on the randomization sequence generation and allocation concealment would improve transparency. Minor grammatical issues are present but do not detract from methodological rigor.

Results

The Results section presents the findings in a structured order and uses appropriate statistical methods. The within-group and between-group comparisons are clearly reported with valid p-values and consistent formatting (mean ± SD and median [range] where appropriate). Significant improvements in tricuspid regurgitation pressure gradient (TRPG) and exercise tolerance in the PCSO-524 group are appropriately highlighted. However, the section would benefit from improved language clarity and consistency in terminology (e.g., repeated use of “visit” vs. “day”). Grammatical errors and typographical mistakes (e.g., “rricuspid” instead of “tricuspid”) should be corrected. The use of vague language such as “numerically different improvement” should be revised to more standard phrasing like “a non-significant trend.” Overall, the statistical validity is strong, but presentation quality needs refinement.

Discussion and Conclusion

The Discussion section provides a broad and generally balanced interpretation of the study’s findings, particularly the observed improvements in exercise tolerance and coughing among dogs receiving PCSO-524. The authors appropriately contextualize their results within the existing literature and acknowledge key limitations, including small sample size, re-examination delays, and the absence of dietary control for omega-3 intake. These admissions strengthen the scientific integrity of the discussion.

However, the section is significantly weakened by numerous grammatical errors, awkward phrasing, and lack of narrative focus. Several paragraphs contain redundancies or imprecise statements (e.g., “may could have biased”), and the flow between ideas is inconsistent. Additionally, some claims are repeated unnecessarily or not fully supported by detailed reasoning (e.g., comparisons of TRPG reductions). References to human studies are overused in places without drawing direct relevance to the canine population. A major language edit is required to improve clarity, eliminate repetition, and present a more coherent, professional narrative.

The Conclusion, while consistent with the overall findings, is underdeveloped. It lacks clarity, contains grammatical issues, and does not adequately emphasize the statistically significant clinical improvements observed in the PCSO-524 group. Furthermore, the final sentence regarding the variability of TRPG is valid but ends the manuscript on a vague note. The conclusion should be rewritten to provide a concise, confident summary of findings and a clear recommendation for future research.

Overall, while the interpretation of results is scientifically reasonable, a thorough rewrite of the Discussion and Conclusion sections is necessary to ensure clarity, accuracy, and proper scholarly tone.

Reviewer #4: This study is well-executed and provides valuable insights into adjunctive therapy for canine PH. Minor improvements in methodological transparency and discussion depth would enhance its impact.

Reviewer #5: no additional comments .

Reviewer #6: Dear Authors, This is a good manuscript and deserves to be published, however there are a few areas for improvement before the manuscript can be accepted.

1. Sample Size is too small. I suggest to label the title to reflect that aspect. You can call it a pilot study or something like that to justify the small sample.

2. The diet is not controlled or recorded. This could significantly influence the study outcomes as it is particularly important when evaluating a nutraceutical intervention. I suggest you acknowledge this more explicitly and, if possible, include retrospective dietary assessments.

3. Study Group 1 (PCSO-524) had significantly worse baseline coughing scores. While this may partially explain the observed improvement, it could also bias the results. I suggest that you consider conducting an ANCOVA adjusting for baseline coughing scores.

4. There is a basic need for defining and expressing the lack of objective and functional endpoints. As the study relies heavily on owner-reported outcomes, it makes it even more important to use more objective data as that will for sure strengthen the findings and improve the quality of the study. You can also discuss this limitation and how further research studies can address it.

5. Kindly elaborate on the implications of the lack of change in NT-proBNP concentrations.

6. Please elaborate the discussion on the composition and other synergistic actions of PCSO-524.

7. Ensure all abbreviations are defined at first mention in both the abstract and main text.

8. Ensure that grammatical errors are corrected before submission and that the English is up to the standards of the journal and general scientific literature.

Reviewer #7: This is a well written and interesting paper. It has great clinical application. There is so little available for this disease so it good to see some research on a natural product that will be helpful for most patients.

The statistical analysis appears to be correct and appropriate.

The data set presented is complete.

The English is appropriately written.

The references contain all the appropriate literature and does include both human and veterinary literature.

On line 273 results is spelled incorrectly.

Reviewer #8: The manuscript presents a prospective, randomized, double-blind, placebo-controlled study assessing the efficacy of Perna canaliculus oil (PCSO-524) as an add-on therapy to sildenafil in dogs with pulmonary hypertension (PH). The subject is novel and relevant, particularly as interest in nutraceuticals grows. The methodology is generally appropriate, and the manuscript is well structured. However, several issues—particularly related to small sample size, lack of dietary control, and interpretation of clinical significance—should be addressed before consideration for publication.

2. Methodology:

Design: Randomized, placebo-controlled design is appropriate.

Blinding: Adequately blinded (owners, investigators, and sponsors), which strengthens the credibility of the findings.

Sample Size: Only 17 dogs completed the study, which severely limits statistical power.

Inclusion Criteria: Well-defined, but would benefit from clearer exclusion criteria (e.g., how arthritis was ruled out).

Nutritional Confounding: No documentation of dietary intake or background EPA/DHA content, which critically affects interpretation of omega-3 related outcomes.

Suggestions for Revision:

Include a table summarizing baseline diets or clearly state the limitation due to dietary variability.

Clarify how orthopedic comorbidities were ruled out (as they may influence exercise tolerance).

Include a table summarizing baseline diets or clearly state the limitation due to dietary variability.

3. Statistical Analysis:

Suggestion: Provide confidence intervals alongside p-values to support effect estimates.

5. Interpretation and Discussion:

The discussion is thorough and references relevant literature.

However, the role of confounding variables (e.g., underlying osteoarthritis, nutrition) is under-emphasized.

The statement that PCSO-524 improved coughing and exercise tolerance should be softened to reflect the exploratory nature of this pilot study.

Suggestion:

Revise tone to reflect preliminary nature and acknowledge limitations more explicitly.

Consider recommending replication in a larger, multicenter cohort.

Final Recommendation:

Major Revision

Summary of Required Revisions:

Address Small Sample Size Limitations in the Discussion and interpret findings with appropriate caution.

Include or Discuss Diet Control and how this may have influenced omega-3 intake.

Clarify and Justify Exclusion of Orthopedic Comorbidities or acknowledge their potential confounding effects.

Correct Minor Language Errors and Typos.

Soften Conclusions to reflect the exploratory nature of the findings.

**Do you want your identity to be public for this peer review?** For information about this choice, including consent withdrawal, please see our Privacy Policy

Reviewer #1: **Yes: ** Hussein Mussa Muafa

Reviewer #2: **Yes: ** SEYED AMIRHOSSEIN MAZHARI

Reviewer #3: No

Reviewer #4: No

Reviewer #5: No

Reviewer #6: **Yes: ** Kevin Morris

Reviewer #7: No

Reviewer #8: No

---

## [Author Response · Author response to Decision Letter 1]

2 Jul 2025

Please see all responses in the file name "response to reviewers".

---

## [Decision Letter · Decision Letter 1]

6 Aug 2025

Dear Dr. Sakcamduang,

Thank you for submitting your manuscript to PLOS ONE. After careful consideration, we feel that it has merit but does not fully meet PLOS ONE’s publication criteria as it currently stands. Therefore, we invite you to submit a revised version of the manuscript that addresses the points raised during the review process.

We look forward to receiving your revised manuscript.

Kind regards,

Doa'a G. F. Al-u'datt

Academic Editor

PLOS ONE

Journal Requirements:

**Additional Editor Comments:**

see attached file

Reviewers' comments:

Reviewer's Responses to Questions

**Comments to the Author**

Reviewer #2: (No Response)

Reviewer #3: All comments have been addressed

Reviewer #6: All comments have been addressed

2. Is the manuscript technically sound, and do the data support the conclusions?

Reviewer #2: Yes

Reviewer #3: Yes

Reviewer #6: Yes

3. Has the statistical analysis been performed appropriately and rigorously?

Reviewer #2: Yes

Reviewer #3: Yes

Reviewer #6: Yes

4. Have the authors made all data underlying the findings in their manuscript fully available?

Reviewer #2: Yes

Reviewer #3: Yes

Reviewer #6: Yes

5. Is the manuscript presented in an intelligible fashion and written in standard English?

Reviewer #2: Yes

Reviewer #3: No

Reviewer #6: Yes

Reviewer #2: The specific questions and recommendations are as follows:

1:

The manuscript would benefit from a structured abstract.

The revised version has the same unstructured abstract as the primary submission.

2:

It is recommended to crate a section for the Ethical considerations. Under Ethical considerations

section, it is required to provide more information in the standard format.

It is better presented in the revised version.

3:

I recommend creating a new section for the study limitations.

Has not been addressed by authors.

4:

It is advised that a new section explaining any acronyms used in the article be added at the end,

following the conclusion section.

No action by authors.

Thanks to the authors for their hard work on this Revision. It’s a pleasure to review such a

contribution to the field.

Reviewer #3: (No Response)

Reviewer #6: (No Response)

**Do you want your identity to be public for this peer review?** For information about this choice, including consent withdrawal, please see our Privacy Policy

Reviewer #2: **Yes: ** SEYED AMIRHOSSEIN MAZHARI

Reviewer #3: No

Reviewer #6: **Yes: ** Kevin Morris

---

## [Author Response · Author response to Decision Letter 2]

14 Aug 2025

Journal Requirements:

Response: We confirm that we have adhered to all journal requirements for the reference list:

• We have reviewed our reference list to ensure it is complete and correct.

• We have verified that no cited articles have been retracted.

• All changes to the reference list made during this revision are noted in this rebuttal letter."

We have modified author contribution (add Nattapon Riengvirodkij to formal analysis, and Walasinee Sakcamduang to writing original draft).

We have also undertaken further revisions to enhance the manuscript. The tables within the main document (Tables 1-4) have been modified for improved clarity and accuracy. Furthermore, the supplementary tables have been revised and are submitted as a separate file in accordance with journal guidelines.

Additional Editor Comments:

5. Is the manuscript presented in an intelligible fashion and written in standard English?

Reviewer #2: Yes

Reviewer #3: No

Reviewer #6: Yes

Response: We are grateful to the reviewers for their comments. In light of the feedback from Reviewer #3, we have performed a careful and thorough proofread of the entire manuscript. We have meticulously corrected all typographical and grammatical errors and revised sentences for clarity and precision. We believe the revised version now fully meets the journal's language requirements.

Point-by-point response to each reviewer comment

Response for Reviewer #2 (file attached ‘Review 55a.pdf’)

1. The manuscript would benefit from a structured abstract. The revised version has the same unstructured abstract as the primary submission.

Response: We thank the reviewer for this suggestion. Per the journal's author guidelines, which specify a single, non-structured paragraph for the abstract, we have maintained this format. We have completely written the abstract to improve its clarity, structure, and flow, ensuring the objective, methods, results, and conclusions are presented more clearly within the required format.

2. It is recommended to crate a section for the Ethical considerations. Under Ethical considerations section, it is required to provide more information in the standard format.

It is better presented in the revised version.

Response: Thank you very much.

3. I recommend creating a new section for the study limitations. Has not been addressed by authors.

Response: We appreciate the reviewer's suggestion to ensure the study limitations are clearly visible. According to standard manuscript structure and PLOS ONE's guidelines, the limitations of a study are addressed within the 'Discussion' section. We have included a detailed paragraph dedicated to these limitations in our revised Discussion, ensuring it is clearly delineated and easy for the reader to identify. We believe this placement follows the journal's required format.

4. It is advised that a new section explaining any acronyms used in the article be added at the end, following the conclusion section. No action by authors.

Response: We thank the reviewer for this advice. In line with PLOS ONE's style guide, we have opted not to include a separate list of abbreviations. Instead, we have carefully reviewed the entire manuscript to ensure that every acronym is defined in full upon its first use in both the abstract and the main text (e.g., 'Pulmonary Hypertension (PH)'). We believe this standard approach adheres to the journal's formatting requirements and ensures clarity for the reader throughout the text.

Thanks to the authors for their hard work on this Revision. It’s a pleasure to review such a

contribution to the field.

Response: Thank you very much for your positive response.

Walasinee Sakcamduang

---

## [Decision Letter · Decision Letter 2]

16 Sep 2025

Clinical improvement in canine pulmonary hypertension with Perna canaliculus oil (PCSO-524) add-on therapy: effects on exercise tolerance and cough

PONE-D-25-20629R2

Dear Dr. Sakcamduang,

We’re pleased to inform you that your manuscript has been judged scientifically suitable for publication and will be formally accepted for publication once it meets all outstanding technical requirements.

Kind regards,

Doa'a G. F. Al-u'datt

Academic Editor

PLOS ONE

Additional Editor Comments (optional):

Reviewer #2:

Reviewers' comments:

Reviewer's Responses to Questions

**Comments to the Author**

Reviewer #2: All comments have been addressed

2. Is the manuscript technically sound, and do the data support the conclusions?

Reviewer #2: Yes

3. Has the statistical analysis been performed appropriately and rigorously?

Reviewer #2: I Don't Know

4. Have the authors made all data underlying the findings in their manuscript fully available?

Reviewer #2: Yes

5. Is the manuscript presented in an intelligible fashion and written in standard English?

Reviewer #2: Yes

Reviewer #2: Thanks to the authors for their hard work on this Revision. It’s a pleasure to review such a

contribution to the field.

**Do you want your identity to be public for this peer review?** For information about this choice, including consent withdrawal, please see our Privacy Policy

Reviewer #2: **Yes: ** SEYED AMIRHOSSEIN MAZHARI

---

## [Editor Report · Acceptance letter]

PONE-D-25-20629R2

PLOS ONE

Dear Dr. Sakcamduang,

I'm pleased to inform you that your manuscript has been deemed suitable for publication in PLOS ONE. Congratulations! Your manuscript is now being handed over to our production team.

Kind regards,

on behalf of

Dr. Doa'a G. F. Al-u'datt

Academic Editor

PLOS ONE